# VISION AS LORA

## ABSTRACT

We introduce Vision as LoRA (VoRA), a novel paradigm for transforming an LLM into an MLLM. Unlike prevalent MLLM architectures that rely on external vision modules for vision encoding, VoRA internalizes visual capabilities by integrating vision-specific LoRA layers directly into the LLM. This design allows the added parameters to be seamlessly merged into the LLM during inference, eliminating structural complexity and minimizing computational overhead. Moreover, inheriting the LLM's ability of handling flexible context, VoRA can process inputs at arbitrary resolutions.

To further strengthen VoRA's visual capabilities, we introduce a block-wise distillation method that transfers visual priors from a pre-trained ViT into the LoRA layers, effectively accelerating training by injecting visual knowledge. Additionally, we apply bi-directional attention masks to better capture the context information of an image. We successfully demonstrate that with additional pre-training data, VoRA can perform comparably with conventional encode-based MLLMs. All training data, codes, and model weights will be released.

## 1 INTRODUCTION

Multimodal Large Language Models (MLLMs) (Li et al., 2023a; Liu et al., 2023; 2024a; Alayrac et al., 2022; Zhu et al., 2023) have advanced significantly by integrating pre-trained vision models with Large Language Models (LLMs) (Chen et al., 2024a; Touvron et al., 2023; Zheng et al., 2023; Brown et al., 2020; Bai et al., 2023) through a modular design: visual features extracted by vision encoders to be aligned with LLMs via a connector, as shown in Figure 1(a). While efficient in training, this approach has key limitations derived from the external vision expert models, i.e., extra computational costs and image resolution constraints. For instance, many vision encoders, particularly Vision Transformers (ViTs) (Zhai et al., 2023; Radford et al., 2021; Dosovitskiy et al., 2020), adhere to a fixed-resolution training paradigm, limiting flexibility. Additionally, the modular design imposes a sequential workflow: the LLM cannot begin processing until the vision encoder and connector have fully processed the image. To overcome these issues, recent studies (RohanBavishi & Taşırlar, 2023; Diao et al., 2025a) have explored unified, encoder-free architectures that process raw pixels directly within a single Transformer (i.e., an LLM), eliminating the need of external vision models. However, such methods face challenges from modality conflicts between vision and language, which would lead to new problems, such as unstable training and catastrophic forgetting issues.

Relevant research (Diao et al., 2025b; Luo et al., 2024) has made attempts to address modality conflicts through

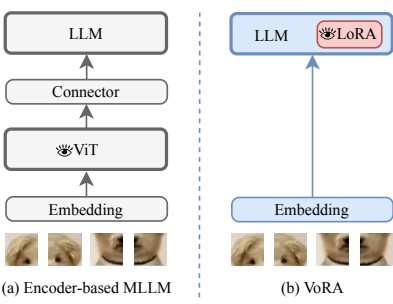

(a) Encoder-based MLLM          (b) VoRA

Figure 1: Visual parameters are indicated with an eye icon. Mainstream MLLMs adopt a modular, sequential architecture: raw pixels are first processed by a pre-trained vision encoder to extract high-level visual features, which are then aligned with the LLM through a modality connector for vision-language tasks. In contrast, VoRA consists solely of an LLM and a lightweight embedding layer. The LoRA layers serve as visual parameters that can be integrated into the LLM without incurring additional computational costs or memory burdens.

parameter decoupling methods. For example, Mono-InternVL (Luo et al., 2024) introduced a Mixture-of-Experts (MoE) framework (Shazeer et al., 2017), employing separate expert modules for vision and language processing. Taking a step further, EVEv2 (Diao et al., 2025b) decoupled all linear and normalization layers in the LLM. While these approaches helped mitigate modality conflicts, they doubled the LLM's parameters, complicating the architecture and substantially increasing memory overhead.

To address these challenges, we propose Vision as LoRA (VoRA), a method of transforming LLMs into encoder-free MLLMs by integrating vision understanding abilities through Low-Rank Adaptation (LoRA) (Hu et al., 2022). While we acknowledge that decoupling vision and language parameters is critical, we wish to avoid dependency on parameter expansion in inference. To this end, VoRA applies trainable LoRA layers to LLMs, which encode the new modality, i.e., vision, while preserving the language knowledge of the original LLM by freezing its parameters, as shown in Figure 1(b). Unlike previous approaches (Diao et al., 2025b; Luo et al., 2024) that retain vision-specific parameters during inference, VoRA merges LoRA layers into the LLM after training, incurring near-zero additional computational cost or memory overhead.

Furthermore, VoRA leverages pre-trained vision models as teacher models to inject visual priors into the LoRA layers. Specifically, we adopt the strategy of block-wise distillation (Hinton et al., 2015): the intermediate visual representations of each LLM block are forced to align with the corresponding block-level features extracted by the teacher model. With such a process, we can greatly accelerate training and reduce the demand for massive data.

In addition, we replace the LLM's causal attention mask with a bi-directional one for image processing, which better captures contextual relations. Meanwhile, we have also found that, unlike most conventional encoder-based MLLMs (Bai et al., 2023; Liu et al., 2023; 2024a; Li et al., 2024a; Zhu et al., 2023; Chen et al., 2023; Wang et al., 2024c) which are constrained by fixed-resolution vision encoders, VoRA naturally supports native image resolutions by exploiting the LLM's inherent ability to process variable-length sequences.

Our contributions are threefold:

- **Framework innovation:** VoRA converts LLMs into MLLMs via: (1) vision as LoRA, (2) block-wise distillation, and (3) bi-directional attention for vision. Parameter decoupling between vision and language pathways stabilizes training, while other components accelerate training and reduce data needs. Ablation studies confirm the effectiveness of each element, establishing VoRA as a new paradigm for encoder-free MLLMs.

- **Performance validation:** When trained with a proper scale of additional data, VoRA matches conventional encoder-based MLLMs in terms of performance while reducing computational costs, demonstrating that LLMs can acquire native multimodal capabilities without external vision models. This challenges the widely perceived necessity of encoder-based architectures for multimodal tasks.

- **Potential extensibility:** Although we narrow down our scope to vision understanding tasks in this paper, the modality-agnostic architecture of VoRA has the potential of generalizing to other modalities (e.g., audio and point clouds) and tasks (e.g., image generation).

## 2 RELATED WORKS

### 2.1 ENCODER-BASED MLLMS

The dominant architecture of MLLMs has remained largely unchanged since its inception, comprising three components: a ViT (Radford et al., 2021; Zhai et al., 2023; Fini et al., 2024), an LLM (Touvron et al., 2023; Brown et al., 2020; Yang et al., 2024; Zheng et al., 2023), and a connector to bridge modality gaps. Previous research has focused primarily on connector design, ranging from simple MLP layers (Liu et al., 2023; 2024a; Zhu et al., 2023; Chen et al., 2023) to hierarchical feature fusion modules (Alayrac et al., 2022; Team, 2024b) or other complex architectures (Wang et al., 2024b;a; Chen et al., 2024b; Tong et al., 2024). Despite these innovations, fundamental limitations persist due to their reliance on external vision encoders. First, computational and memory overhead escalates dramatically when applying multiple vision encoders (Tong et al., 2024) or scaling to larger ones (Wang et al., 2024c). Second, fixed-resolution pre-training of ViTs forces MLLMs to

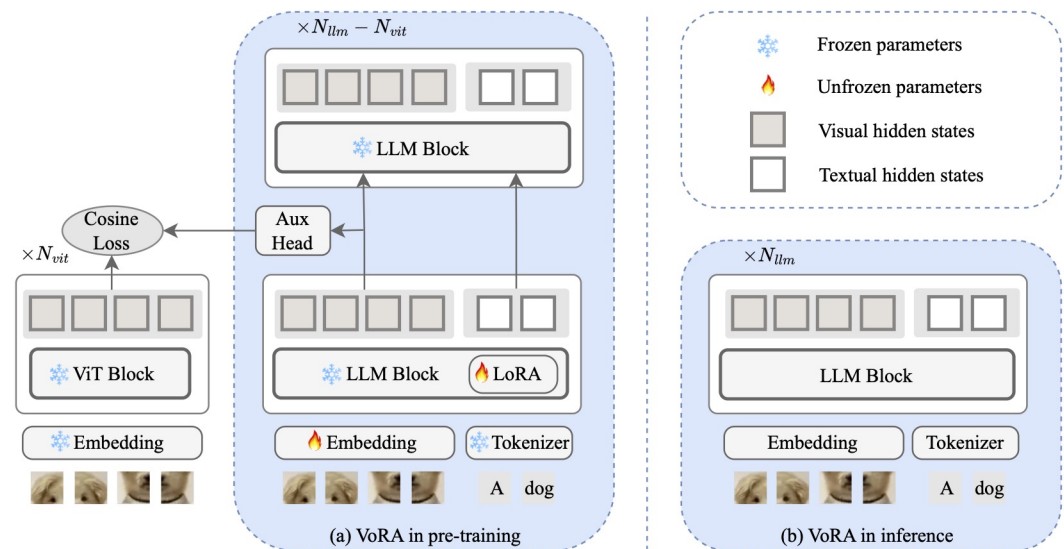

Figure 2: The architecture of VoRA. Figure (a) shows the architecture of VoRA in pre-training: in this stage, VoRA only unfreezes the LoRA layers for vision and the visual embedding layer, i.e., a shallow MLP layer with a positional embedding. Figure (b) shows VoRA in inference: the LoRA layers are merged into the LLM, and thus the only added parameters are a shallow embedding layer (about 6M parameters).

employ workarounds like image tiling (Liu et al., 2024a; Li et al., 2024a) or restricted square resolutions (Wang et al., 2024c; Bai et al., 2023). Recent attempts (Agrawal et al., 2024; Wang et al.; Fini et al., 2024) to train resolution-agnostic ViTs have remained impractical, in that they adopted massive proprietary data and opaque training procedures. These challenges have spurred interest in encoder-free architectures that could bypass ViTs entirely.

## 2.2 ENCODER-FREE MLLMS

The pioneering work, Fuyu (RohanBavishi & Taşırlar, 2023), demonstrated the feasibility of training encoder-free models on interleaved image-text data, though at prohibitive computational costs with limited technical transparency. Subsequent approaches, such as EVE (Diao et al., 2025a), reduced the vision encoder parameters to a single Transformer block, aligning its output features with a ViT through distillation while updating all LLM parameters to learn about vision during the main training stage. However, these methods still struggle with conflicts between the LLM's inherent language abilities and the new modality, i.e., vision. These conflicts arise from the coupled language and vision parameters, which exacerbate unstable training and lead to catastrophic forgetting of the original language abilities.

To overcome these problems, Mono-InternVL (Luo et al., 2024) and EVEv2 (Diao et al., 2025b) proposed parameter decoupling strategies inspired by the MoE method (Shazeer et al., 2017), duplicating LLM parameters for vision-specific processing while freezing its original weights. Despite successfully addressing forgetting issues and modality conflicts, these methods suffered from substantial memory overhead by doubling model parameters, compromising architectural simplicity. Our work addresses this by applying LoRA, which encodes vision while maintaining the language abilities of the LLM, and can be merged into the LLM without causing additional memory overhead.

## 3 VISION AS LORA

In this section, we introduce three key components of VoRA: vision as LoRA, block-wise distillation, and bi-directional attention masks for vision.

### 3.1 Stabilize training: Vision as LoRA

As shown in Figure 2(a), we integrate LoRA layers into the LLM to enable vision understanding. During pre-training, images are first converted into vision embeddings using a lightweight embedding layer, i.e., a shallow MLP with positional encodings of about 6M parameters. Let $N_{\text{vit}}$ and $N_{\text{llm}}$ denote the number of blocks in the ViT and the LLM, respectively. We apply LoRA to all linear layers within the first $N_{\text{vit}}$ blocks of the LLM, including query-key-value (QKV) projections and feed-forward network (FFN) layers. Crucially, only the LoRA parameters and the vision embedding layer are updated during training, while the original LLM parameters remain frozen. This design decouples vision and language parameters, stabilizing training compared to full LLM training and avoiding the training collapse observed in prior works (Diao et al., 2025a).

Figure 2(b) demonstrates that after pre-training, the LoRA parameters can be seamlessly merged into the base LLM, thereby eliminating additional inference overhead.

### 3.2 Boost training: block-wise distillation

We introduce a block-wise distillation paradigm to align VoRA's intermediate visual representations with the block-wise features of a pre-trained ViT. This approach transfers visual knowledge from the ViT via knowledge distillation (Hinton et al., 2015; Fang et al., 2023), accelerating training while reducing dependence on large-scale vision data. Unlike conventional distillation that updates entire models, we only update the vision-specific LoRA layers within the LLM. Specifically, for each block $i$ in the first $N_{\text{vit}}$ layers of the LLM, we align its hidden states with those of block $i$ in the ViT. The training objective combines the following two components.

**Distillation loss.** For each transformer block $i$ and vision token position $s$, we maximize cosine similarity between projected LLM features and ViT embeddings via:

$$\mathcal{L}_{\text{distill}}^{i} = \frac{1}{S} \sum_{s=1}^{S} \left( 1 - \frac{\text{AuxHead}(\boldsymbol{h}_{\text{llm}}^{i,s})^{\top} \boldsymbol{h}_{\text{vit}}^{i,s}}{\|\text{AuxHead}(\boldsymbol{h}_{\text{llm}}^{i,s})\|_2 \|\boldsymbol{h}_{\text{vit}}^{i,s}\|_2} \right), \tag{1}$$

where $S$ is the ViT's output sequence length (number of vision embeddings to represent one image), $\boldsymbol{h}_{\text{llm}}^{i,s}, \boldsymbol{h}_{\text{vit}}^{i,s} \in \mathbb{R}^M$ denote the hidden states for the $s$-th token in block $i$, and $\text{AuxHead}(\cdot)$ is a projection layer (RMSNorm (Zhang & Sennrich, 2019) + linear layer) adapting LLM features to the ViT's embedding space. The loss is averaged across $N_{\text{vit}}$ blocks:

$$\mathcal{L}_{\text{distill}} = \frac{1}{N_{\text{vit}}} \sum_{i=1}^{N_{\text{vit}}} \mathcal{L}_{\text{distill}}^{i}. \tag{2}$$

**Language modeling loss.** For image-caption pairs, we optimize caption generation using cross-entropy, which is consistent with the standard approach used in LLMs:

$$\mathcal{L}_{\text{LM}} = - \sum_{t=t_0}^{T} \log P(w_t | w_{<t}, \boldsymbol{x}_{\text{image}}), \tag{3}$$

where $T$ is the total sequence length, $\boldsymbol{x}_{\text{image}}$ represents vision inputs, and $t_0$ indexes the first caption token.

**Final objective.** The final loss combines both objectives:

$$\mathcal{L}_{\text{total}} = \mathcal{L}_{\text{distill}} + \mathcal{L}_{\text{LM}}. \tag{4}$$

### 3.3 Bi-directional attention masks for vision

While bi-directional attention masks is common in Transformer architectures in various fields (Dosovitskiy et al., 2020; Radford et al., 2021; Zhou et al., 2024), few studies have explored replacing the causal mask of autoregressive LLMs with a bi-directional mask, especially in the field of MLLMs.

As illustrated in Figure 3 , we have explored the use of a bi-directional attention mask for vision. Our findings indicate that this attention mask positively impacts the final performance of VoRA, which will be discussed in Section 5. In contrast to prior works (Diao et al., 2025a;b; Luo

| Data Format | Dataset | # Sample | Total |
|---|---|---|---|
| Image Caption | DataComp29M-recap (ours) | 29M | 30.4M |
| | GLDv2-recap (ours) | 1.4M | |
| Text QA | Infinity-Instruct-3M (BAAI, 2024) | 3.5M | 6.4M |
| | SmolTalk (Allal et al., 2025) | 1.0M | |
| | OpenOrca (Lian et al., 2023) | 994.0K | |
| | MathInstruct (Xiang Yue, 2023) | 262.0K | |
| | OrcaMath (Mitra et al., 2024) | 200.0K | |
| | MagpiePro (L3 ST) (Li et al., 2024a) | 150.0K | |
| | WizardCoder (Luo et al., 2023) | 143.0K | |
| | OpenCodeInterpreter (Zheng et al., 2024) | 66.0K | |
| | MathQA (Amini et al., 2019) | 29.8K | |
| | Dolly (Conover et al., 2023) | 11.0K | |

Table 1: Data used in the pre-training stage of VoRA. We use a mixture of both image and text data to alleviate the forgetting issue in training.

et al., 2024; RohanBavishi & Taşırlar, 2023), which have relied on causal masking designed for autoregressive text generation, we demonstrate that adopting bi-directional attention for vision tokens while retaining causal masking for text, not only preserves language capabilities but also enhances visual performance. This aligns with insights from image generation research (Zhou et al., 2024), highlighting VoRA's potential as a unified architecture for multimodal generation and understanding tasks.

As shown in Figure 3, we explored three types of attention masks for vision: (a) causal mask, (b) bidirectional mask, and (c) localized bidirectional mask. While the bidirectional mask demonstrates improved performance, we find that the localized bidirectional mask outperforms it by allowing tokens to focus exclusively on a single image without interference from other text.

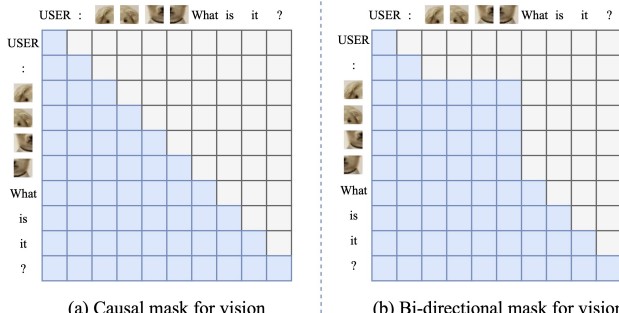

(a) Causal mask for vision     (b) Bi-directional mask for vision

## 4 DATA

### 4.1 DATA COLLECTION AND PREPROCESSING

We claim that the primary focus of this work is not on data engineering or filtration; therefore, we adopt a straightforward data collection and processing strategy. Following previous studies (Diao et al., 2025a;b; Luo et al., 2024), our pre-training framework utilized re-captioned

Figure 3: Attention masks for vision: (a) causal attention inherits the autoregressive mask from language modeling, enforcing sequential dependency between image patches; (b) bidirectional attention offers full visibility between all image patches within the same input, enabling global contextual awareness.

data. Given the limited availability of open-source, large-scale re-captioned datasets, we employed Qwen2-VL-72B (Wang et al.) to generate captions for images sampled from DataComp-1B (Gadre et al., 2023). From this raw dataset, we selected approximately 29 million images with a longer edge exceeding 448 pixels.

We recognize that this dataset lacks specific world knowledge, particularly regarding landmarks, celebrities, and artworks. To address the deficiency in landmark data, we supplemented our dataset with approximately 1.4 million images from the Google Landmarks Dataset v2 (GLDv2) (Weyand

et al., 2020). For other categories, no suitable million-scale datasets were available. Furthermore, due to potential ethical concerns, we chose not to collect such data. Consequently, we acknowledge that our method may underperform in these domains. However, this limitation can be mitigated in future works by integrating relevant datasets.

## 4.2 MULTIMODAL DATA MIXTURE

While VoRA decouples vision and language parameters, we have observed that extended caption-only training slightly degrades the LLM's instruction-following capability. To preserve this ability, we mixed text instruction data into the training data. As shown in Table 1, our final mixture contained approximately 30M image-caption pairs and 6.4M text instruction samples. The text data were obtained directly from: Infinity-Instruction (BAAI, 2024), SmolTalk (Allal et al., 2025), Cambrian-1 (Tong et al., 2024), and LLaVA-OneVison (Li et al., 2024a).

## 5 EXPERIMENTS

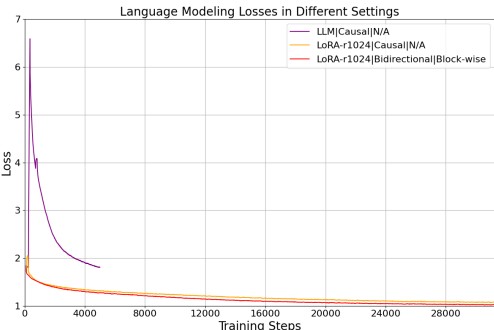

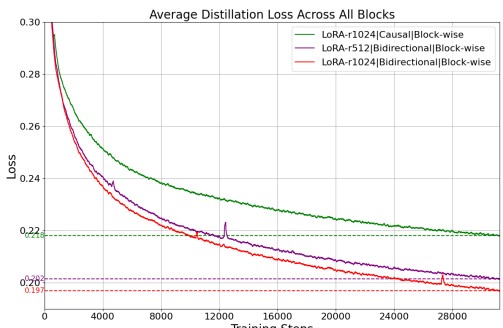

Figure 4: Language modeling losses in different settings. Training the full LLM with a new modality of data can lead to unrecoverable spike in loss curve, i.e., loss collapse.

Figure 5: Average distillation loss across all blocks under various settings. Our LoRA-r1024|Bidirectional|Block-wise configuration achieves the lowest average distillation loss across all blocks. This indicates a closer alignment with the ViT's feature space, confirming that bi-directional attention masks and a larger rank of LoRA layers also enhance visual knowledge transfer.

### 5.1 IMPLEMENTATION DETAILS

**Training setup.** Unless otherwise specified, we employed AIMv2-Huge-448p (Fini et al., 2024) as the default vision encoder and Qwen2.5-7B-Instruct (Yang et al., 2024) as the LLM across all experiments. The pre-training learning rate was fixed at 0.0002 (held constant unless explicitly varied), with 100 warm-up steps and a global batch size maintained at 256. All other hyperparameters and optimizer configurations followed the defaults in (Liu et al., 2024a).

For fine-tuning, all LoRA layers were merged into the LLM, while other components (e.g., distillation modules) were eliminated. The full LLM and 6M-parameter visual embedding layer were trainable. For native-resolution variants (VoRA-AnyRes in Table 2), we retained the pre-trained weights of the fixed-resolution version and adopted native-resolution strategy only during fine-tuning.

**Benchmarks.** As shown in Table 2 and Table 3, we evaluated the model on several benchmarks: VQAv2: VQAv2 (Goyal et al., 2017); SQA-I: ScienceQA-Image (Lu et al., 2022); TQA: TextVQA (Singh et al., 2019); POPE: POPE (Li et al., 2023b); $\text{MMP}_\text{p}$: MME Perception (Fu et al., 2023); $\text{MME}_\text{c}$: MME Cognition (Fu et al., 2023); MMB: MMBench (Liu et al., 2024b); SEED-I: SEED-Image (Li et al., 2024b); MMVet: MMVet (Yu et al., 2023); AI2D: AI2D (Kembhavi et al., 2016); RQA: Realworld-QA (Team, 2024a); MMMU: MMMU (Yue et al., 2024).

## 5.2 ABLATION STUDIES

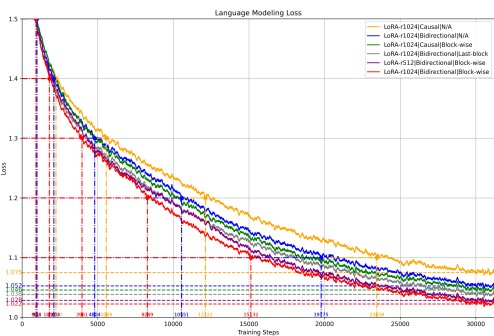

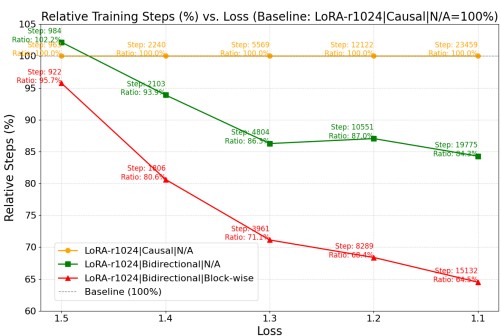

Figure 6: Pre-training loss curves under different configurations. Loss values are smoothed (window=100) for visual clarity. The data sampling order was fixed to ensure fair comparison, as evidenced by the similar trajectories of the loss curves in various settings. LoRA-r1024|Bidirectional|Block-wise refers to the setting: LoRA with rank 1024, bi-directional attention masks for vision, and block-wise distillation. The configuration with the lowest loss was adopted as the default setting in our experiments.

Figure 7: Data efficiency analysis. Our experiments demonstrate that combining bidirectional attention masks for vision tokens with block-wise knowledge distillation significantly improves data efficiency compared to the vanilla LoRA configuration. Furthermore, as the target loss decreases (e.g., from 1.5 to 1.1), the required data proportion relative to the baseline diminishes progressively, indicating higher data efficiency.

| Vision Params | Visual Att. Mask | Distill. type | TQA | POPE | $MME_p$ | MMB | SEED-I | MMVet | AI2D | RQA | Avg. |
|---|---|---|---|---|---|---|---|---|---|---|---|
| LoRA-r1024 (2B) | Causal | N.A. | 43.7 | 78.6 | 1137.7 | 47.7 | 57.8 | 20.6 | 49.9 | 49.7 | 50.6 |
| LoRA-r1024 (2B) | Bidirectional | N.A. | 43.6 | 80.9 | 1132.8 | 49.1 | 58.7 | 17.9 | 47.2 | 51.5 | 50.7 |
| LoRA-r1024 (2B) | Causal | Block-wise | 45.1 | 82.7 | 1172.9 | 52.9 | 63.7 | 20.1 | 50.9 | 51.2 | 53.2 |
| LoRA-r1024 (2B) | Bidirectional | Last-block | 44.6 | 82.5 | 1197.5 | 51.8 | 63.8 | 17.9 | 49.9 | 52.8 | 52.9 |
| LoRA-r512 (1B) | Bidirectional | Block-wise | 47.2 | 83.3 | 1280.5 | 57.6 | 65.3 | 18.5 | 55.9 | 53.1 | 55.6 |
| LoRA-r1024 (2B) | Bidirectional | Block-wise | 50.1 | 83.8 | 1224.5 | 53.7 | 65.1 | 22.8 | 52.1 | 55.8 | 55.6 |

Table 2: The performance of various settings on standard benchmarks reveals that lower loss during pre-training correlates with better performance. "LoRA-r1024 (2B)" indicates that the rank for the LoRA layers is set to 1024, with approximately 2 billion parameters unfrozen for training in total.

Our ablation studies focused on three key components of VoRA: vision as LoRA, block-wise distillation, and bi-directional attention masks for vision. We employed two primary methods to assess performance in various settings: the pre-training loss on an 8M subset of our DataComp29M-recap dataset, as illustrated in Figure 6, and metrics from eight benchmarks, presented in Table 2. Additionally, we visualized the average distillation loss across all blocks, as shown in Figure 5.

**Ablation on vision as LoRA.** Training the full-parameter LLM proved unstable due to modality conflicts (Figure 4), consistent with findings in (Diao et al., 2025a). While reducing the learning rate to a lower value allowed us to observe successful training cases, the loss decreased more slowly than that of LoRA-1024. Therefore, we have excluded it from our primary experiments.

Next, we analyzed different LoRA rank configurations in VoRA. Figure 6 shows that a rank of 512 resulted in a slightly higher loss (+0.006) compared to rank 1024. This trend continued in the distillation loss (Figure 5), where rank 512 showed a modestly higher average block-wise distillation loss (+0.005) compared to rank 1024. Although both configurations ended up with the same average score of 55.6 (Table 2), the consistent loss advantage suggested that higher ranks might have better optimization potential. Furthermore, we experienced training instability with rank 1536, which

prompted us to choose rank 1024 as the default configuration.

**Ablation on bi-directional attention masks.** As demonstrated in Figure 6, under fixed hyperparameters (e.g., LoRA rank and distillation type), the bi-directional attention mask consistently achieved lower training loss compared to causal masking. This empirical advantage was further supported by the reduced average distillation loss across all Transformer blocks, as depicted in Figure 5. Quantitatively, as evidenced in Table 2, replacing causal masking with bi-directional masks yielded significant performance improvements. For instance, switching from LoRA-r1024|Causal|Block-wise to LoRA-r1024|Bidirectional|Block-wise led to a 2.4-point average score gain, while replacing LoRA-r1024|Causal|N/A with LoRA-r1024|Bidirectional|N/A yielded a gain of 0.1 points.

**Block-wise distillation.** As shown in Figure 6 and Table 2, applying distillation to the final Transformer block alone significantly improved training efficiency. For example, the transition from the configuration LoRA-r1024|Bidirectional|N/A to LoRA-r1024|Bidirectional| Last-block yielded a 2.7-point score gain and a 0.016 reduction in loss. Extending distillation to all blocks via block-wise supervision further enhanced performance: compared with LoRA-r1024|Bidirectional|Last-block, LoRA-r1024|Bidirectional|Block-wise produced an additional 2.7-point gain and 0.016 loss reduction. These results indicated that the vanilla distillation method, i.e., last-block distillation, could accelerate training, and block-wise distillation could even strengthen this effect.

**Data efficiency analysis.** We measured data efficiency by reporting the relative number of training steps required to reach certain loss thresholds, using vanilla LoRA as the baseline. As illustrated in Figure 7, the bi-directional attention variant without distillation (LoRA-r1024|Bidirectional|N/A) required 102.2% of the baseline training steps to reach Loss=1.5, whereas adding block-wise distillation (LoRA-r1024|Bidirectional|Block-wise) reduced this to 95.7%. The efficiency gap became more pronounced at lower loss: at Loss=1.1, the same configurations needed 84.3% and 64.5% of the vanilla LoRA baseline steps, respectively. This demonstrated that our optimal configuration achieved equivalent convergence with 35.5% fewer training steps than vanilla LoRA.

Furthermore, the ratio of data needed by our best configuration relative to vanilla LoRA decreased over time, implying that comparable performance could be achieved with $N \times$ fewer training data.

## 5.3 STANDARD EVALUATION

To ensure a fair comparison between VoRA and existing methods, we deliberately restricted our experimental design. While prior works (e.g., EVE, EVEv2 (Diao et al., 2025b), and Mono-InternVL (Luo et al., 2024)) have leveraged massive in-domain datasets (Table 3), such approaches complicated direct comparisons due to proprietary training data. Our goal is not to pursue state-of-the-art performance on benchmarks but to validate a novel MLLM architecture. Thus, we limited fine-tuning to the publicly available LLaVA-665K dataset without additional scaling.

To eliminate the potential advantages provided by LLMs and ViTs, we also trained a LLaVA-1.5 model using Qwen-2.5-7B and AIMv2-0.6B. As shown in Table 3, prior encoder-free methods often adopted intricate multi-stage pipelines involving module freezing strategies and proprietary datasets (e.g., 100M–1.2B samples). In contrast, our framework employed a streamlined single-stage training process (pre-training followed by fine-tuning), using about 30M image-text pairs.

As shown in Table 3, VoRA achieved performance comparable to both official and reproduced LLaVA-1.5 baselines on most benchmarks when evaluated under strict LLaVA-1.5

| Method | Posters | Celebrity | Landmark | Artwork | Total |
|---|---|---|---|---|---|
| LLaVA-1.5 | 156.1 | 143.5 | 173.5 | 134.0 | 607.1 |
| VoRA | 117.3 | 111.2 | 139.3 | 105.5 | 473.3 |
| VoRA-AnyRes | 110.2 | 104.7 | 138.0 | 110.8 | 463.7 |

Table 4: The performance of VoRA in world knowledge tasks. We acknowledge its deficiency, as expected, due to the lack of relevant in-domain data in our pre-training dataset. This is the primary reason for our lower performance on the MME Perception benchmark.

protocols (Liu et al., 2024a), i.e., identical prompts/generation parameters. However, VoRA underperformed on MME Perception, a gap we attribute to limited world knowledge in our pre-training data. This was further quantified in Table 4, where VoRA struggled with tasks demanding intensive world-knowledge: 1) inferring movie details from posters, 2) identifying celebrities, 3) recognizing

| Method | LLM | ViT | # Sample | | VQAv2 | SQA-I | TQA | POPE | $MME_p$ | $MME_c$ | MMB | SEED-I | MMVet | AI2D | RQA | MMMU |
|---|---|---|---|---|---|---|---|---|---|---|---|---|---|---|---|---|
| | | | Pretrain | Finetune | | | | | | | | | | | | |
| *Encoder-based* | | | | | | | | | | | | | | | | |
| BLIP2 | Vicuna-13B | EVA-1B | 129M | - | 65.0 | 61 | 42.5 | 85.3 | 1293.8 | - | - | 49.7 | 22.4 | - | - | - |
| InstructBLIP | Vicuna-7B | EVA-1B | 129M | 1.2M | - | 60.5 | 50.1 | - | - | - | 36 | 58.8 | 26.2 | - | - | - |
| InstructBLIP | Vicuna-13B | EVA-1B | 129M | 1.2M | - | 63.1 | 50.7 | 78.9 | 1212.8 | - | - | - | 25.6 | - | - | - |
| LLaVA-1.5 | Vicuna-7B | CLIP-0.3B | 558K | 665K | 78.5 | 66.8 | 58.2 | 85.9 | 1510.7 | 316.1 | 64.3 | 66.1 | 31.1 | 54.8 | 54.8 | 35.3 |
| LLaVA-1.5 | Qwen2.5-7B | AIMv2-0.6B | 558K | 665K | 82.3 | 77.5 | 59.2 | 85.2 | 1582.3 | 313.0 | 66.3 | 70.6 | 33.7 | 63.7 | 60.0 | 35.3 |
| *Encoder-free* | | | | | | | | | | | | | | | | |
| EVE | Vicuna-7B | ~~CLIP-0.3B~~ | 49M(2) | 665K | 75.4 | 63.0 | 51.9 | 83.6 | 1217.3 | 266 | 49.5 | 61.3 | 25.6 | 48.5 | - | - |
| EVE-HD | Vicuna-7B | ~~CLIP-0.3B~~ | 49M(2) | 1.8M | 74.2 | 64.9 | 56.8 | 85.0 | 1305.7 | 322 | 52.3 | 64.6 | 25.7 | 61.0 | - | - |
| EVEv2 | Qwen2.5-7B | - | 87M(2) | 665K | - | 72 | 57 | - | - | - | - | - | - | - | - | - |
| Mono-InternVL | Intern1.5-2B | - | 922M | 665K | - | 57 | 49 | - | 1100 | - | - | - | - | 42 | - | - |
| Mono-InternVL | Intern1.5-2B | - | 1.2B(2) | 665K | - | 58 | 55 | - | 1110 | - | - | - | - | 46 | - | - |
| VoRA | Qwen2.5-7B | ~~AIMv2-0.6B~~ | 30M | 665K | 76.0 | 75.9 | 56.3 | 84.5 | 1363.4 | 311.1 | 64.2 | 67.5 | 33.7 | 65.6 | 57.7 | 32.2 |
| VoRA-AnyRes | Qwen2.5-7B | ~~AIMv2-0.6B~~ | 30M | 665K | 76.0 | 72.0 | 58.7 | 85.5 | 1336.1 | 319.3 | 61.3 | 68.9 | 33.7 | 61.1 | 60.1 | 32.0 |

Table 3: Comparison with previous methods on several benchmarks. Since this paper aims to demonstrate that VoRA is a strong base model, we did not scale the fine-tuning data. Therefore, we did not compare with recent state-of-the-art models that often require additional data engineering or involve proprietary datasets; methods that utilize extra fine-tuning data are grayed out. We classified domain-specific VQA data as fine-tuning data rather than pre-training data for EVEv2 and Mono-InternVL, which differs from their original classification in the respective papers. The notation "49M(2)" indicates that this method employs a two-stage training process using a total of 49M image-text pairs. The strikethrough notation ~~ViT~~ means that ViT is excluded during inference.

landmarks, and 4) classifying artworks, as these tasks required external domain knowledge absent in our training datasets.

# 6 LIMITATIONS

The most significant limitation of VoRA lies in its reliance on additional pre-training data to compensate for the absence of an external vision model, because the LLM has to learn visual feature extraction from scratch. While we hypothesize that scaling VoRA could surpass encoder-based MLLMs by avoiding information loss in the pre-trained ViT (as theorized in (Diao et al., 2025a; Tong et al., 2024)), we currently lack the empirical evidence to confirm this advantage. Limited training data and computational resources have prevented us from observing a clear performance crossover point. We leave this promising hypothesis for future exploration.

# 7 CONCLUSION

VoRA establishes a new paradigm for converting LLMs into MLLMs through three components: (1) vision as LoRA, (2) Block-wise distillation, and (3) bi-directional attention masks for vision. By integrating vision capabilities directly into the LLM via mergeable LoRA layers for visual encoding, VoRA eliminates the need for a separate vision model. This unified approach reduces memory overhead, lowers computational costs, and leverages the LLM's inherent flexibility in context length to process native-resolution images with minimal adaptation. This design bypasses the problems brought by using ViT as an external vision model while still decoupling the vision and language parameters to ensure stable training.

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
