# OpenReview forum: "Vision as LoRA"
_ICLR.cc/2026/Conference — ICLR 2026 Conference Withdrawn Submission_

### Official Review · Reviewer_o7LA · 2025-10-24

**Soundness:** 2
**Presentation:** 2
**Contribution:** 2
**Rating:** 2
**Confidence:** 4

**Summary:**

This work proposes a novel monolithic large vision-language model based on embedding the visual modality inside an LLM through lower-rank adaptation layers. Coupled with this architecture, the work also proposes a new alignment training stage for aligning the earlier LLM blocks' representations with a visual encoder. Through experimental evaluation on several vision-language benchmarks the work aims to highlight the strengths of its proposed methodology.

**Strengths:**

The primary strengths of the work could be listed as the following:

- The broad idea of integrating a new modality (in this case vision) into an LLM through lower-rank adaptation layers is an interesting idea to explore, and has not been explored in the similar manner in the literature before.
- Experimental results demonstrate the effectiveness of the method over contemporary monolithic baselines, as evidenced by largely positive improvements in Table 3.
- Overall idea is communicated relatively clearly and the methodology is easy to grasp and follow.

The minor strengths of the work could be listed as the following:

- The authors of the work provided several honest notes on their design choices, such as the data curation aspect or reproducing LLaVA 1.5 baseline with a stronger visual encoder and an LLM, which is also appreciated.

**Weaknesses:**

**W1 - Computational Overhead and Added Parameters:** In several parts of the work, there are repeated claims of "reduced computational burdens and memory requirements". However, there are no comparisons against the baselines, neither with EVE variants [A, B], Monolithic Intern-VL [C] or with encoder-decoder VLMs [D, E, F] in terms of FLOPs, memory occupancy or overall training and inference costs. These comparisons are particularly important as the work claims that other approaches double the number of parameters [A, B, C] and VoRA fixes these issues, although it is not very clear to which extent VoRA does so. Particularly, it can be seen that the *"low-rank"* layers introduced to accommodate the vision modality are applied to **all linear layers with unconventionally high bottleneck ranks of 512 and 1024, resulting in a 1B/2B** increase in parameters. Given EVE [A] includes a much stricter parameter sharing mechanism compared to VoRA and EVEv2 [B] only disentangles the FFNs for different modalities, it is not clear whether this work achieves a better efficiency/performance trade-off.

**W2 - Ambiguous Representations in Tables:** Table 3 shows decent improvements for VoRA over other monolithic baselines. However, there is an ambiguity in the tables: While the visual encoder parameters are explicitly mentioned for encoder-decoder methods, the additionally introduced _"low-rank layer parameters"_ are not mentioned for VoRA. Given that the method introduces anywhere from 1B additional parameters to 2B, and that it very clearly underperforms the existing encoder-decoder baselines (LLaVA 1.5 variants versus VoRA in Table 3), it is not clear what the advantage of the proposed method is over the well-established encoder-decoder baselines.

**W3 - Bidirectional Attention Presented as a Novelty:** The type of bidirectional attention masking for visual modality is a well-known practice in the Large VLM literature [E, F], though it is presented as a novelty in the work. Although its application to monolithic VLMs may be novel, it is not proposed by this work initially and should not be framed as such.

**W4 - Missing Details and Ablations on LoRA:** There is no mention of the $\alpha$ parameter used for LoRA. Furthermore, there are no ablations with smaller LoRA ranks, such as $16, 32, 64$, which are the common practice in the field [G, H]. Finally, there are no ablations with adapting only the Q and V projection matrices with LoRA instead of applying it to every single linear projection, which was found to strike a better efficiency/performance trade-off in the original LoRA work [G].

Below are the relatively minor weaknesses of the work:

- Some of the tables have formatting issues, such as Tables 2 and 3, making it really hard to figure out the benchmark names. Improving them would greatly improve the readability of the results.
- The work brings in additional training costs, just like the other monolithic baselines it considers. At least a small comparison on the total GPU/TPU hours on training VoRA versus the EVE variants or the encoder-decoder baselines such as LLaVA 1.5 would be great to have.

---
[A] Diao, H., Cui, Y., Li, X., Wang, Y., Lu, H., & Wang, X. (2024). Unveiling encoder-free vision-language models. Advances in Neural Information Processing Systems, 37, 52545-52567.

[B] Diao, H., Li, X., Cui, Y., Wang, Y., Deng, H., Pan, T., ... & Wang, X. (2025). Evev2: Improved baselines for encoder-free vision-language models. arXiv preprint arXiv:2502.06788.

[C] Luo, G., Yang, X., Dou, W., Wang, Z., Liu, J., Dai, J., ... & Zhu, X. (2025). Mono-internvl: Pushing the boundaries of monolithic multimodal large language models with endogenous visual pre-training. In Proceedings of the Computer Vision and Pattern Recognition Conference (pp. 24960-24971).

[D] Liu, H., Li, C., Li, Y., & Lee, Y. J. (2024). Improved baselines with visual instruction tuning. In Proceedings of the IEEE/CVF conference on computer vision and pattern recognition (pp. 26296-26306).

[E] Beyer, L., Steiner, A., Pinto, A. S., Kolesnikov, A., Wang, X., Salz, D., ... & Zhai, X. (2024). Paligemma: A versatile 3b vlm for transfer. arXiv preprint arXiv:2407.07726.

[F] Steiner, A., Pinto, A. S., Tschannen, M., Keysers, D., Wang, X., Bitton, Y., ... & Zhai, X. (2024). Paligemma 2: A family of versatile vlms for transfer. arXiv preprint arXiv:2412.03555.

[G] Hu, E. J., Shen, Y., Wallis, P., Allen-Zhu, Z., Li, Y., Wang, S., ... & Chen, W. (2022). Lora: Low-rank adaptation of large language models. ICLR, 1(2), 3.

[H] Dettmers, T., Pagnoni, A., Holtzman, A., & Zettlemoyer, L. (2023). Qlora: Efficient finetuning of quantized llms. Advances in neural information processing systems, 36, 10088-10115.

**Questions:**

- Can you comment on what the performance would look like with smaller LoRA ranks ($16, 32, 64$) and/or with only applying LoRA to Q and V projection matrices as it is the common practice for LoRA-adapting LLMs?

- Can you detail the computational resources utilized for the work and compare it against the baselines, both in terms of training (GPU/TPU hours) and inference costs (FLOPs, memory requirements)?

- Can you discuss the overall stand of VoRA compared to LLaVA 1.5 and other encoder-decoder variants? I understand the drawbacks of the modality-specific modular variants (e.g. different stages of training, other costs), but from what I can tell, VoRA suffers from similar issues (there are 1B/2B more parameters to accommodate vision, which is much larger than the visual encoder of LLaVA 1.5, and VoRA also necessitates an additional stage of training for aligning the vision-specific parameters)?

- Can you explain what you mean by "merging the LoRA parameters"? Is this just keeping the trained LoRA layers in-tact as they normally were during training while performing inference? If so, how is this approach more efficient than the encoder-decoder baselines?

---

### Official Review · Reviewer_WKpy · 2025-10-31

**Soundness:** 2
**Presentation:** 3
**Contribution:** 3
**Rating:** 2
**Confidence:** 4

**Summary:**

The paper proposed to integrate LoRa layers into an LLM to endow it with visual understanding without the need of a vision encoder. The method, called VoRa, is based on applying LoRA to layers within as many blocks of the LLM as there are blocks in a teacher ViT. The authors explored multiple variations of VoRa, including the rank, attention type and distillation techniques for boosting the performance of the new LoRa parameters.

**Strengths:**

The proposed method is addressing an important topic of reducing the computational costs of training new VLMs

**Weaknesses:**

- Limited empirical evaluation on both visual and textual tasks.
- VoRa still relies on a pretrained ViT for good performance via distillation.

**Questions:**

In my opinion, this work is lacking empirical evidence that would strongly support the benefits of the introduced VoRa method. In particular:
- As seen from Table 1 and Figure 5, a lower loss doesn’t always correspond to a better performance: for example, bidirectional attention in Figure 5 has a lower loss but does not yield better results in Table 2 (row 1 vs 2).
- VoRa still depends on a pretrained vision encoder for a good performance as seen in the results when comparing row 1 and row 3 in Table 1 with the results in Table 3.
- Causal vs bidirectional attention only makes a difference when you distil, as seen in comparing row 1->2  vs row 3-> 6 in Table 2.
- VoRA-AnyRes performs worse than VoRA on 5 out of 12 benchmarks presented in Table 3, which makes its effectiveness questionable.
- No evidence that VoRA doesn’t lead to catastrophic forgetting on tasks requiring textual understanding and image captioning.

A suggestion for the authors on how to improve the experimental evaluation would be to:
- include experiments on image captioning as well tasks requiring textual understanding to validate that catastrophic forgetting doesn’t happen with VoRa.
- Ablation on the influence of the vision teacher.
- Expanding the results with information on the number of extra parameters, memory requirements etc, especially when comparing to existing methods.
- Validating the stability of the chosen LoRa rank across architectures, especially because the authors themselves acknowledged that VoRa sufferers from the rank instabilities.

---

### Official Review · Reviewer_P8Hm · 2025-11-01

**Soundness:** 3
**Presentation:** 2
**Contribution:** 3
**Rating:** 4
**Confidence:** 3

**Summary:**

This paper introduces a novel training and architectural paradigm that integrates visual capability into the LoRA modules of LLM layers, called Vision as LoRA (VoRA). To further enhance visual understanding, the authors employ block-wise distillation and bi-directional attention for the vision modality. Experimental results demonstrate that the VoRA-trained MLLM achieves performance comparable to encoder-based MLLMs and outperforms encoder-free MLLMs.

**Strengths:**

- This paper introduces a novel paradigm, VoRA, which transforms an LLM into an MLLM without using a ViT. The idea is genuinely interesting and innovative, and it represents one of the most efficient approaches for building lightweight MLLMs.

- Although block-wise distillation and bi-directional attention for the vision modality are not new techniques, the authors effectively demonstrate their usefulness in this context.

**Weaknesses:**

- The idea itself is interesting, and as shown in Table 3, VoRA outperforms existing encoder-free-based models, demonstrating its effectiveness. However, its performance is still about 6% lower than that of encoder-based models (e.g., LLaVA-1.5-7B) on benchmarks such as VQA. Personally, I expected VoRA to be more efficient even if the performance were slightly lower or comparable. However, considering that (1) the pre-training dataset size is much smaller (558K vs. 30M) and (2) the number of parameters seems to be roughly similar (approximately comparable to AIMv2-0.6B), it makes me wonder whether simply fine-tuning an existing MLLM with LoRA or pre-training an existing MLLM architecture might actually be more effective. I would like to hear the authors’ thoughts on this point.

- As a follow-up, to convincingly demonstrate the significance of VoRA, I believe it would be important to include a comparison of inference speed with models like LLaVA-1.5-7B, as well as an evaluation of whether VoRA is truly flexible to any image resolution. Although Table 4 briefly addresses this, additional experiments under diverse resolution settings would help substantiate the authors’ claims more strongly.

- Regarding block distillation, since the number of ViT layers and LLM layers differ, how was the distillation performed? Was it based on the early layers of the LLM corresponding to the ViT layers? The paper does not describe this clearly. If the authors indeed used the initial layers as the reference, I assume the motivation was to prevent degradation of the LLM’s inherent verbalization ability (i.e., its original linguistic knowledge) that might occur if distillation were applied to the last layers. I am curious if this assumption is correct. I understand that additional experiments may be difficult during the rebuttal period, but a brief explanation or clarification would be appreciated.

Below are several minor suggestions to improve readability (these do not affect the review score):
- L244–245: What does “(c) localized bidirectional mask” refer to? It does not appear in Figure 3.
- The font size of the loss plots is too small and difficult to read. Please unify it with the main text font size.
- In Tables 2 and 3, the spacing between dataset names is too tight; this can be easily adjusted in LaTeX.
- In Table 4, the “Total” column appears to overflow.
- L069–079: The description of experimental results is too brief and generic (e.g., “our method performs well”). It would be more informative to specify concrete numbers, such as “our method outperforms encoder-free-based model A by an average of +6% across six benchmarks.”

**Questions:**

See Weaknesses.

---

### Official Review · Reviewer_eNw3 · 2025-11-01

**Soundness:** 3
**Presentation:** 2
**Contribution:** 2
**Rating:** 6
**Confidence:** 3

**Summary:**

VoRA is an encoder-free multimodal LLM that internalizes vision by inserting LoRA adapters into the first (N) Transformer blocks of a base LLM (covering QKV and FFN) and training only these adapters plus a ~6M-parameter visual embedding layer, then merging the adapters back for near-zero-overhead inference. Training is stabilized and made data-efficient via block-wise distillation from a pretrained ViT and bidirectional (including localized) attention masks over image tokens. Pretrained on ~30M image–text pairs and 6.4M text-only instructions,

**Strengths:**

1. Architectural simplicity & inference efficiency. Visual competence is encoded via mergeable LoRA, avoiding a separate vision encoder at inference; only a small embedding layer remains to be added.

2. Stabilization & transfer mechanisms. Block-wise distillation aligns intermediate LLM states with ViT features; bi-directional masks for vision reduce training loss and improve scores versus causal masking.

3. Clear positioning vs. prior encoder-free lines. The paper articulates how LoRA-based decoupling avoids the parameter doubling used by MoE-style decoupling while still mitigating forgetting.

4. Any-resolution fine-tuning path. By eschewing a fixed-resolution ViT, the method can adopt a native-resolution strategy at fine-tune time.

**Weaknesses:**

1. Underperformance on world-knowledge perception. VoRA trails on MME-Perception and on tasks involving posters/celebrities/landmarks/artworks; authors attribute this to data composition.

2. Training resource transparency. Key reproducibility metadata (GPU type/count, GPU-hours) are not reported.

3. Limited evidence on language integrity post-merge. The paper mixes 6.4M text instructions to protect instruction following, but provides no dedicated pure-text benchmarks (e.g., MMLU, GSM8K) to quantify any drift after merging.

**Questions:**

1. Language ability after the merge. Report MT-Bench/MMLU/GSM8K (or equivalent) before vs. after LoRA merge to demonstrate no degradation in pure-text performance. What, if any, drift is observed?

2. Teacher sensitivity. How sensitive is training stability and final performance to the choice/size of the teacher ViT? Any results with alternative teachers?

3. World-knowledge gap. With a small, licensed in-domain supplement (e.g., 10^5 - 5*10^5 pairs) for posters/celebrities/artworks, how much of the MME-Perception gap closes without harming generalization?

---

### Note · Authors · 2026-01-06

I have read and agree with the venue's withdrawal policy on behalf of myself and my co-authors.